

# Sea cucumbers reduce chromophoric dissolved organic matter in aquaculture tanks

Seyed Mohammad Sadeghi-Nassaj[1], Teresa S. Catalá[1], Pedro A. Álvarez[2] and Isabel Reche[1]

[1] Departamento de Ecología and Instituto del Agua, Facultad de Ciencias, Universidad de Granada, Granada, Spain
[2] iMare Natural S.L., Motril, Granada, Spain

## ABSTRACT

**Background**. Mono-specific aquaculture effluents contain high concentrations of nutrients and organic matter, which affect negatively the water quality of the recipient ecosystems. A fundamental feature of water quality is its transparency. The fraction of dissolved organic matter that absorbs light is named chromophoric dissolved organic matter (CDOM). A sustainable alternative to mono-specific aquaculture is the multitrophic aquaculture that includes species trophically complementary named "extractive" species that uptake the waste byproducts. Sea cucumbers are recognized as efficient extractive species due to the consumption of particulate organic matter (POM). However, the effects of sea cucumbers on CDOM are still unknown.

**Methods**. During more than one year, we monitored CDOM in two big-volume tanks with different trophic structure. One of the tanks (−holothurian) only contained around 810 individuals of *Anemonia sulcata*, whereas the other tank (+holothurian) also included 90 individuals of *Holothuria tubulosa* and *Holothuria forskali*. We routinely analyzed CDOM absorption spectra and determined quantitative (absorption coefficients at 325 nm) and qualitative (spectral slopes) optical parameters in the inlet waters, within the tanks, and in their corresponding effluents. To confirm the time-series results, we also performed three experiments. Each experiment consisted of two treatments: +holothurians (+H) and −holothurians (−H). We set up three +H tanks with 80 individuals of *A. sulcata* and 10 individuals of *H. tubulosa* in each tank and four −H tanks that contained only 80 individuals of *A. sulcata*.

**Results**. In the time-series, absorption coefficients at 325 nm ($a_{325}$) and spectral slopes from 275 to 295 nm ($S_{275-295}$) were significantly lower in the effluent of the +holothurian tank (average: $0.33\ \mathrm{m^{-1}}$ and $16\ \mathrm{\mu m^{-1}}$, respectively) than in the effluent of the −holothurian tank (average: $0.69\ \mathrm{m^{-1}}$ and $34\ \mathrm{\mu m^{-1}}$, respectively), the former being similar to those found in the inlet waters (average: $0.32\ \mathrm{m^{-1}}$ and $22\ \mathrm{\mu m^{-1}}$, respectively). This reduction in the absorption of the dissolved organic matter appears to be mediated by the POM consumption by holothurians. The experiments confirmed the results observed in the time-series. The $a_{325}$ and $S_{275-295}$ values were significantly lower in the treatment with holothurians than in the treatment without holothurians indicating a reduction in the concentration of chromophoric organic compounds, particularly of low molecular weight.

**Discussion**. Consequently, sea cucumbers appear to improve water transparency in aquaculture tanks. The underlying mechanism of this improvement might be related

Corresponding author
Isabel Reche, ireche@ugr.es

to the POM consumption by holothurians, which reduces the concentration of CDOM derived from POM disaggregation or to the direct assimilation of dissolved compounds of low molecular weight as chromophoric amino acids.

# INTRODUCTION

The exponential growth of human population has boosted the global demand of fish and seafood (*FAO, 2009*). Nevertheless, the extractive fisheries are more and more reduced and the aquaculture is gaining importance accounting for more than 40% of human consumption of fish and seafood (*Bostock et al., 2010*). Mono-specific aquaculture produces wastewater that usually contains high concentrations of organic matter as well as inorganic nutrients, antibiotics and uneaten food pellets (*Read & Fernandes, 2003*; *Klinger & Naylor, 2012*). At ecosystem level, the effluents of mineral nutrients associated with the aquaculture activity can produce problems of eutrophication (*Ajin et al., 2016*; *Ruiz-Zarzuela et al., 2009*). On the other hand, the loads of dissolved and particulate organic matter with the effluents can reduce water transparency due to an increase in light backscattering and absorption (*Ibarra, Cembella & Grant, 2012*; *Del Bel Belluz et al., 2016*). Therefore, a sustainable aquaculture with effluents of low environmental impact is a global challenge for both scientists and food producers. The polyculture and the integrated multitrophic aquaculture (IMTA) is an alternative practice to alleviate the handicaps of the traditional, mono-specific aquaculture (*Diana et al., 2013*). Unlike mono-specific aquaculture, polyculture and IMTA uses trophically complementary "extractive" species that consume the excretion products, fecal and food wastes of the primary species reducing these loads in the effluents (*Chopin et al., 2012*). Hence, it is desirable that the future expansion of aquaculture promotes this practice to reduce the inputs of organic matter in the environment, at the same time that aquaculture farmers can obtain an economical value from the co-cultured species.

The chromophoric dissolved organic matter (CDOM) is the fraction of the dissolved organic matter (DOM) that absorbs light in the ultraviolet (UV) and, to a lesser extent, in the visible range of the spectrum. Therefore, CDOM is largely responsible for UV and blue light attenuation in marine ecosystems (*Bricaud, Morel & Prieur, 1981*; *Nelson & Siegel, 2013*). Since CDOM absorption overlaps one of the chlorophyll a absorption peaks, CDOM can diminish the potential for primary productivity. This fact also affects the algorithms used in remote sensing to determine ocean color and infer primary productivity (*Carder et al., 1989*; *Siegel et al., 2005*; *Ortega-Retuerta et al., 2010*). Remote sensing has been suggested as an excellent tool to monitor at large scale the impact of offshore aquaculture (*Populus et al., 1995*; *Rajitha, Mukherjee & Chandran, 2007*; *Saitoh et al., 2011*). However, the relation between aquaculture waste and CDOM has been scarcely explored (*Ibarra, Cembella & Grant, 2012*; *Nimptsch et al., 2015*; *Del Bel Belluz et al., 2016*).

Sea cucumbers are highly demanded food for human consumption in some countries (*Purcell et al., 2013*), but they are also important extractive species with a high capacity to consume waste particulate organic matter in sediment deposits (*Nelson, MacDonald & Robinson, 2012a*; *Nelson, MacDonald & Robinson, 2012b*; *Yokoyama, 2013*; *Yokoyama, 2015*). Despite the effects of sea cucumbers on different components of the particulate organic matter has been studied, particularly in open waters under fish cages or mollusk rafts (*Slater & Carton, 2009*; *Slater, Jeffs & Carton, 2009*; *Nelson, MacDonald & Robinson, 2012a*; *Zamora & Jeffs, 2011*; *Yokoyama, 2013*; *Yokoyama, 2015*; *Zhang et al., 2014*), their influence on the optical properties of the dissolved organic matter still remains unexplored (*Zamora et al., 2016*). The influence of sea cucumbers on the optical properties of the organic matter can be relevant in land-based installations submitted to long-term water recirculation.

In this study, we evaluate the effects of sea cucumbers (*Holothuria tubulosa* and *Holothuria forskali*) on optical properties of the dissolved organic matter in aquaculture tanks with *Anemonia sulcata* as primary species. *A. sulcata* is a very palatable species, highly demanded for catering in Spain with also a great pharmacological interest. During one year, we monitored the changes in DOM optical properties in a big-volume (50,000 l) tank with holothurians and in another similar tank without them, exploring the main factors that determine CDOM changes. Subsequently, to corroborate the observations obtained in the time-series of the big-volume tanks we performed three short-term experiments manipulating the presence of holothurians in small tanks (300 l). We observed, both in the time-series of the big tanks and in the short-term experiments, that the presence of holothurians reduced significantly the absorption due to dissolved organic matter increasing, consequently, the water transparency in comparison with the tanks without holothurians. Therefore, holothurians appear to have a high environmental value to improve the water quality in aquaculture installations.

# MATERIAL AND METHODS

## Time-series in the big-volume tanks

We monitored during more than one year two aquaculture tanks located at the iMareNatural S.L. facilities (http://www.imarenatural.com) in Motril, Spain (36°44′38″N, 3°35′59″W). Each tank of 50,000 liters of capacity was connected directly with the coastal water by one inlet pipe (inlet water) and the water from each tank was released by one outlet pipe located in the bottom of the tank (effluent). The seawater was pumped into the tanks at a continuous flow of $1,200 \, l \, h^{-1}$. Therefore, water residence time in the tanks was ca. 42 h and the total annual effluents accounted for 10,512 m$^3$. In one of the tanks, $811 \pm 125$ individuals of the primary species, the sea anemone *A. sulcata*, and $93 \pm 3$ adults of sea cucumbers *Holothuria tubulosa* ($\approx$80%) and *Holothuria forskali* ($\approx$20%) were included (hereafter designated as + *holothurian* tank). In the other tank only $690 \pm 87$ individuals of the primary specie were included (hereafter designated as—*holothurian* tank). Sea anemones were placed on floating plastic boxes in the surface of the tanks and holothurians were free in the bottom and walls of the tanks. Sea anemones were fed

with about 900–1,800 g of fresh chopped fish, mainly *Scomber scombrus* (*Chintiroglou & Koukouras, 1992*) twice per week.

Water samples for different chemical and biological analysis were taken biweekly from July 17th 2013 to August 20th 2014. Each sampling day, we took water samples from the inlet pipe, the center of the two tanks using a telescopic stick with a plastic beaker located in its extreme and from their corresponding effluents. To avoid that light can affect absorption measurements, we immediately took the CDOM samples in pre-combusted (4 h at 500 °C), acid-cleaned, amber glass bottles. They were kept in ice during transportation to the laboratory (about one hour from the tanks). Water samples were filtered through pre-combusted Whatman GF/F glass fiber filters of 0.7 μm nominal pore size and the <0.7 μm fraction was used for the optical characterization of chromophoric dissolved organic matter.

## Chromophoric Dissolved Organic Matter (CDOM)

Absorption spectra of dissolved organic matter provide information on CDOM concentration and other qualitative properties. Absorption coefficients at specific wavelengths (e.g., 325 nm and 443 nm) are used as proxies of CDOM concentration, and the spectral slopes and spectral ratios, which are largely independent of the concentration, are surrogates of CDOM origin, molecular weight and chemical structure (*Weishaar et al., 2003*; *Twardowski et al., 2004*; *Helms et al., 2008*; *Nelson & Siegel, 2013*; *Martínez-Pérez et al., 2017*).

CDOM absorbance spectra were recorded at wavelengths from 200 nm to 750 nm at 1-nm interval using an UV/VIS Perkin Elmer spectrometer with a 10 cm-quartz cuvette. The spectrophotometer was connected to a computer with Lambda 25 software. The detection limit of the spectrophotometer (0.001Absorbance) corresponds to a CDOM absorption coefficient detection limit of 0.02 m$^{-1}$. Spectrum corrections due to residual scattering by fine size particle fractions, micro-air bubbles, or colloidal material present in the sample were performed by subtracting the average of the absorption between 600 and 700 nm (*Green & Blough, 1994*).

CDOM absorption coefficients, $a_\lambda$, were calculated using the next equation:

$$a_\lambda = 2.303 \frac{\text{Absorbance}(\lambda) - \text{Absorbance}(600 - 700)}{l} \tag{1}$$

where $a_\lambda$ is the absorption coefficients in m$^{-1}$ at each $\lambda$ wavelength, *Absorbance* ($\lambda$) is the absorbance at wavelength $\lambda$, *Absorbance (600–700)* is the average absorbance from 600 to 700 nm, 2.303 is the factor that converts decadic to natural logarithms, $l$ is the cuvette path length in m$^{-1}$.

Spectral slopes describe the shape decay of absorption coefficients vs. wavelengths. Slopes were calculated from the linear regression of log-transformed absorption coefficients in the wavelength bands 275–295 nm ($S_{275-295}$) and 350–400 nm ($S_{350-400}$) (*Helms et al., 2008*). The spectral slopes for both wavelength ranges were calculated as in Eq. (2).

$$a_\lambda = a_{\lambda\text{ref}} e^{-S(\lambda - \lambda_{\text{ref}})} \tag{2}$$

Where $\lambda$ is the selected wavelength in nm, $a_\lambda$ is the absorption coefficient at $\lambda$ wavelength in m$^{-1}$, $a_{\lambda\text{ref}}$ is the absorption coefficient at a reference wavelength $\lambda_{\text{ref}}$, and $S$ is the spectral
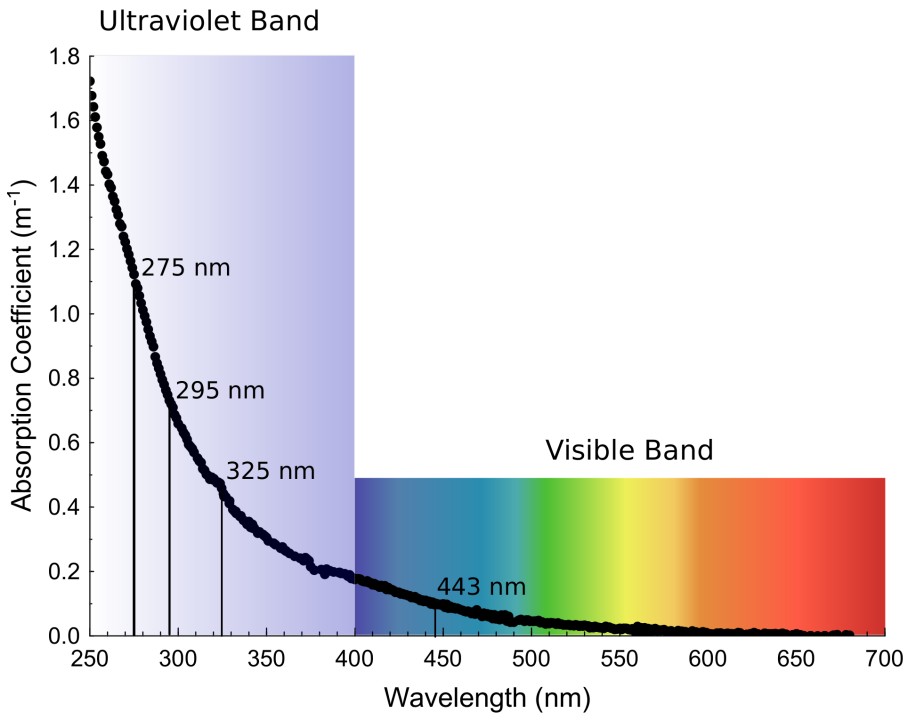

**Figure 1** **Exponential decay of absorption coefficients as wavelengths increase from the ultraviolet to the visible band.** Example of a decay curve of the chromophoric dissolved organic matter in a sample taken on January 30th of 2014 from the big-volume tank that contains holothurians. The wavelengths selected for the calculation of the optical parameters used in this study are marked on the decay curve.

slope. The spectral slope ratio ($S_R$) was calculated as the ratio of the spectral slope from 275 nm to 295 nm ($S_{275-295}$) to the spectral slope from 350 nm to 400 nm (*Helms et al., 2008*).

CDOM absorption usually decreases exponentially as wavelengths increase. Therefore, the shorter the wavelength, the more sensitive to changes is (Fig. 1). Despite a low sensitivity (Fig. 1), the visible wavelength at 443 nm is used in remote sensing studies due to its correspondence with the satellite sensor (*Siegel et al., 2005*; *Ortega-Retuerta et al., 2010*). Therefore, the spectral slope from 275 nm to 295 nm is the most sensitive optical parameter of CDOM changes (*Helms et al., 2008*), but this parameter is not quantitative. CDOM quantity was measured as absorption coefficient at 325 nm ($a_{325}$), since this wavelength is the most common in the literature (*Nelson & Siegel, 2013*; *Catalá et al., 2015*) and has a higher sensitivity than 443 nm (Fig. 1).

### Ancillary data

Basic parameters as temperature (°C), pH, salinity (psu), total dissolved solids (TDS), and conductivity (mScm-1) were measured in the tanks using a multi-parameter HANNA probe (HI9828 model; Woonsocket, RI, USA). Total organic carbon (TOC) concentration was measured as non-purgeable organic carbon after a high-temperature catalytic oxidation using a Shimadzu TOC-V CSN. Samples, by triplicate, were acidified with hydrochloric

acid and purged for 20 min to eliminate the remaining dissolved inorganic carbon. Three to five injections were analyzed for each sample. Standardization of the instrument was done with potassium hydrogen phthalate. Particulate organic matter (POM) was obtained filtering between 1.5 and 2.0 l of water through pre-weighed and pre-combusted (500 °C for 4 h) Whatman GF/F glass fiber filters. The filters containing all the solids were dried at 60 °C for >24 h and reweighed to determine the total mass (mineral + organic matter). Then, the organic fraction was burned by combusting the filters at 500 °C for 6 h; finally, the filters were reweighed again to determine the mineral residue. POM was obtained after the subtraction of the mineral residue to the total mass. The concentration of chlorophyll-*a* was determined spectrophotometrically after pigment extraction with methanol (*APHA, 1992*). Bacteria abundance was determined in triplicate using flow cytometry (*Gasol & Del Giorgio, 2000*) with a FACScalibur Becton Dickinson cytometer equipped with a laser emitting at 488 nm. Data were processed using Cell quest software.

## Short-term experiments

To test the statistical significance of the results obtained in the time-series, we performed three short-term (3 days) experiments. Each experiment was carried out in seven tanks of 300 liters that contained a floating plastic box with 80 individuals of *A. sulcata* per tank and consisted of two treatments: + *holothurians* (+H) and –*holothurians* (−H). At the initial time, in three of the tanks we included 10 individuals of *H. tubulosa* in each tank. These three tanks are the replicates of the +*holothurians* treatment. The other four tanks only contained the 80 individuals of *A. sulcata* and represent the replicates of the –*holothurians* treatment. The experiment 1 was carried out from 6th to 9th October 2017, the experiment 2 from 27th to 30th October 2017, and the experiment 3 from 3rd to 6th November 2017. During the duration of each experiment the anemones were not fed to control the net effect of holothurian activity. At the initial and final time we took samples for the optical characterization of dissolved organic matter. To analyze the samples we followed the same procedures used in the time-series.

## Statistical analysis

To compare the time-series of the CDOM optical parameters in the inlet water with the effluents from the tank with holothurians and the tank without holothurians we performed paired *t*-test (for normally distributed variables) and Wilcoxon matched pairs test (for not-normally distributed variables) using the Statistica software (V8). These statistical analyses ameliorate the problem of temporal pseudoreplication in this type of studies (*Millar & Anderson, 2004*). Correlations between CDOM optical parameters and potential controlling factors were performed using Statistica software (V8). In the short-term experiments to test the statistical significance of the presence of holothurians on the CDOM optical parameters we performed analysis of variance (ANOVA) comparing the tanks with holothurians (+H) with the tanks without holothurians (−H) using Statistica software (V8).

## RESULTS AND DISCUSSION

During the study period, the pH in the inlet waters ranged from 7.71 to 8.31, the temperature from 13.58 °C to 25.58 °C, the salinity from 35.8 to 41.6 psu, the conductivity between 52.28 and 61.96 mS cm$^{-1}$ and total dissolved solids from 18.26 to 30.84 ppt. These basic parameters were similar in the tanks and in their corresponding effluents (Tables S1–S5).

### Time-series of the optical parameters in the big-volume tanks

In the inlet waters, the $a_{325}$ values ranged from 0.06 to 0.83 m$^{-1}$ (Table S1) and in the effluents of +*holothurian* and –*holothurian* tanks from 0.06 to 0.79 m$^{-1}$ and from 0.37 to 1.27 m$^{-1}$, respectively (Tables S2 and S3). The absorption coefficients of the inlet waters (i.e., coastal waters of Western Mediterranean Sea) were similar to those ones found in other coastal waters (*Catalá et al., 2013*; *Nima et al., 2016*) or in the open Mediterranean Sea (*Bracchini et al., 2010*; *Organelli et al., 2014*). Systematically, throughout the time-series, the effluent of the –*holothurian* tank showed higher $a_{325}$ values (grey triangles in Fig. 2A) than the effluent of the + *holothurian* tank (red squares in Fig. 2A). These last values were similar to the $a_{325}$ values in the inlet waters (white circles).

Spectral slopes and spectral slope ratios are qualitative parameters, which are independent of the CDOM concentration. The higher the spectral slope, the smaller the DOM molecular weight is (*Helms et al., 2008*). The slope in the band from 275 nm to 295 nm ($S_{275-295}$) is an optical parameter particularly sensitive to environmental changes as solar radiation or salinity (*Helms et al., 2008*; *Helms et al., 2013*; *Catalá et al., 2013*). In the inlet water, the values of $S_{275-295}$ ranged from 10 to 38 μm$^{-1}$ (Table S1) and in the effluents of +*holothurian* and –*holothurian* tanks from 6 to 28 μm$^{-1}$ and from 13 to 40 μm$^{-1}$, respectively (Tables S2 and S3). In the inlet water, the values were similar to those reported for coastal and estuary waters, usually characterized with lower slopes (∼15–25 μ m$^{-1}$) than the values for the open ocean (∼25–50 μ m$^{-1}$) (*Helms et al., 2008*; *Helms et al., 2013*; *Catalá et al., 2015*). Like the $a_{325}$ values, the $S_{275-295}$ values in the effluent waters of the –*holothurian* tank (grey triangles in Fig. 2B) showed consistently higher values than the inlet water (white circles in Fig. 2B) and in the effluents of the +*holothurian* tank (red squares in Fig. 2B). In the inlet waters, the spectral slope ratios ($S_R$) ranged from 0.6 to 2.6 (Table S1) and in the effluents of +*holothurian* and –*holothurian* tanks from 0.5 to 3.1 and from 0.4 to 3.9, respectively (Tables S2 and S3). The $S_R$ values in –*holothurian* effluents (grey triangles in Fig. 2C) showed consistently higher values than the inlet water (white circles in Fig. 2C) and +*holothurian* effluent (red squares in Fig. 2C).

To assess if the presence of holothurians can modify significantly CDOM we performed paired $t$-test or Wilcoxon matched pair tests pooling all the time-series data (Table 1). In the Fig. 3 we show all the time-series data pooled in median values, 25–75% percentiles and non-outliers values. The $a_{325}$ values in the –*holothurian* tank and effluent (Fig. 3A, grey boxes) were significantly higher that the values in the +*holothurian* tank and effluent and the inlet water (Table 1). A similar effect was found for the spectral slope ($S_{275-29}$) (Fig. 3B, grey boxes), and the spectral ratios ($S_R$) (Fig. 3C, grey boxes). Indeed, we observed higher CDOM concentration with higher spectral slopes (surrogate of smaller molecular size; *Helms et al., 2008*) in the tank without holothurians than in the tank with holothurians.

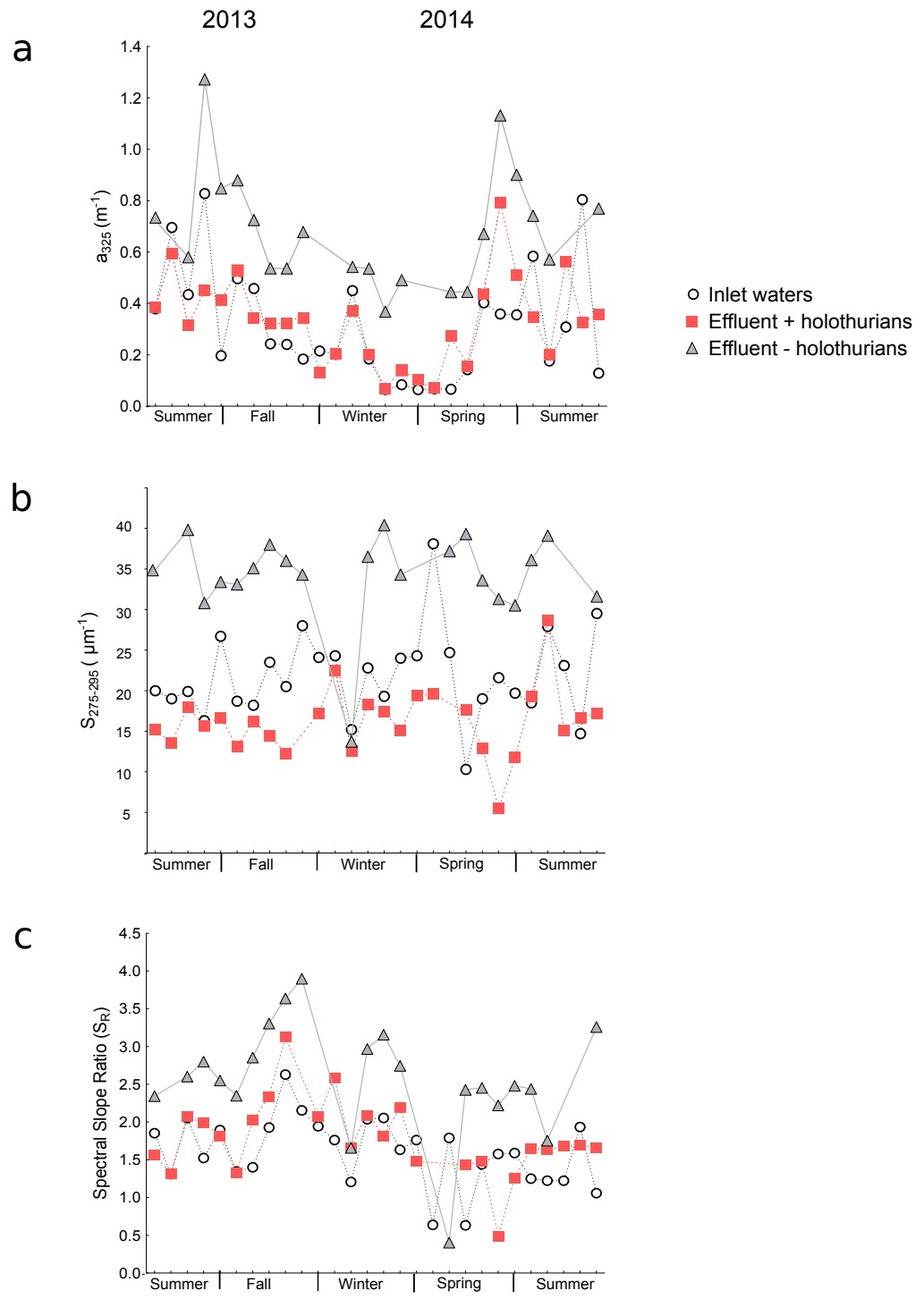

**Figure 2  Time-series of the optical parameters of chromophoric dissolved organic matter in the big-volume tanks.** Values of (A) absorption coefficients at 325 nm ($a_{325}$), (B) spectral slopes from 275 to 295 nm ($S_{275-295}$), and (C) spectral slope ratios ($S_R$) in the inlet water (white circles), in the effluent water of the +*holothurian* tank (red squares) and in the effluent water of the –*holothurian* tank (grey triangles).

**Table 1  Results of paired *t*-test and Wilcoxon matched pairs test between the inlet water and the effluent of the tank with holothurians and the effluent of the tank without holothurians.** Results of paired *t*-test (for normally distributed variables) and Wilcoxon matched pairs test (for not normally distributed variables) between the inlet water and the effluent of the tank with holothurians (+*holothurians*) and the effluent of the tank without holothurians (−*holothurian*) for the CDOM optical properties considered in this study. Bold means that there is a significant difference.

| | Statistical analysis | *t* or *z* | *p*-value |
|---|---|---|---|
| **Inlet water vs. +*holothurian* effluent** | | | |
| Absorption coefficients ($a_{325}$) | Paired *t*-test | 19.67 | **0.0000** |
| Spectral slope ($S_{275-295}$) | Paired *t*-test | 4.86 | **0.0000** |
| Spectral slope ratio ($S_R$) | Wilcoxon | 2.35 | **0.0188** |
| **Inlet water vs. −*holothurian* effluent** | | | |
| Absorption coefficients ($a_{325}$) | Paired *t*-test | 24.91 | **0.0000** |
| Spectral slope ($S_{275-295}$) | Wilcoxon | 3.98 | **0.0000** |
| Spectral slope ratio ($S_R$) | Paired *t*-test | 6.18 | **0.0000** |
| **+*holothurian* vs. −*holothurian* effluents** | | | |
| Absorption coefficients ($a_{325}$) | Paired *t*-test | 11.45 | **0.0000** |
| Spectral slope ($S_{275-295}$) | Wilcoxon | 3.82 | **0.0001** |
| Spectral slope ratio ($S_R$) | Wilcoxon test | 3.78 | **0.0001** |

Therefore, holothurians appear to reduce significantly CDOM concentration, particularly of compounds with comparatively lower molecular weight making the spectral slopes smaller. On the other hand, the differences between the inlet water and the +*holothurian* tank and effluent water, although significant, were less relevant (Fig. 3, Table 1).

These results suggest that effluent of the monoculture of *A. sulcata* increases CDOM in comparison with the inlet water, which could affect the recipient coastal waters. These higher CDOM values in the −*holothurian* tank in comparison with the values in +*holothurian* tank could be related to: (1) a higher abundance of bacteria and their metabolic by-products that produce CDOM or (2) a higher concentration of particulate organic matter (derived from uneaten food, detritus and microbial cells) which disaggregation in dissolved compounds also produce CDOM. In both cases, an increment in CDOM concentration is expected in absence of holothurians. Several studies have shown that bacteria and phytoplankton can produce CDOM as metabolic by-products (*Nelson, Siegel & Michaels, 1998*; *Nelson, Carlson & Steinberg, 2004*; *Ortega-Retuerta et al., 2009*; *Romera-Castillo et al., 2010*; *Catalá et al., 2015*; *Catalá et al., 2016*). However, we did not find significant relationships between the $a_{325}$ values and the concentration of chlorophyll-*a* (+holothurian $r^2 = 0.009$, $p = 0.513$; -holothurian $r^2 = 0.002$, $p = 0.804$) or the abundance of bacteria (+holothurian $r^2 = 0.014$, $p = 0.403$; -holothurian $r^2 = 0.068$, $p = 0.095$) (Fig. 4A). Therefore, phytoplankton and bacterial carbon processing appear to have a minor importance in these tanks. In contrast, we found significant and positive relationships between the concentration of particulate organic matter (POM) and the $a_{325}$ values in the +*holothurian* tank and effluent water (red squares; $r^2 = 0.41$, $p = 0.002$; regression line $a_{325} = 0.20 + 0.079$ POM) and the $a_{325}$ values in the −*holothurian* tank and effluent water (grey triangles; $r^2 = 0.20$, $p = 0.006$; regression line $a_{325} = 0.42 + 0.102$
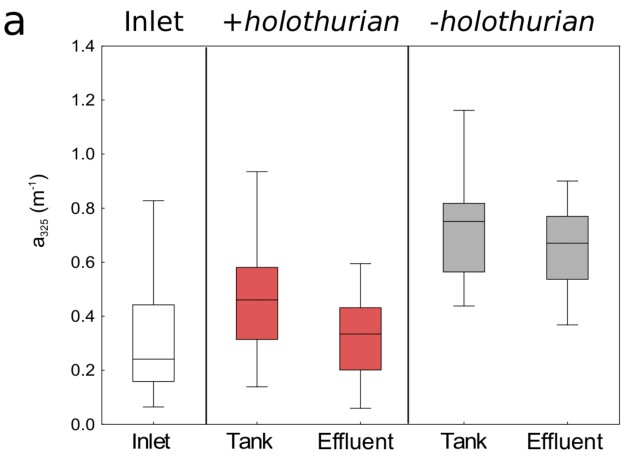

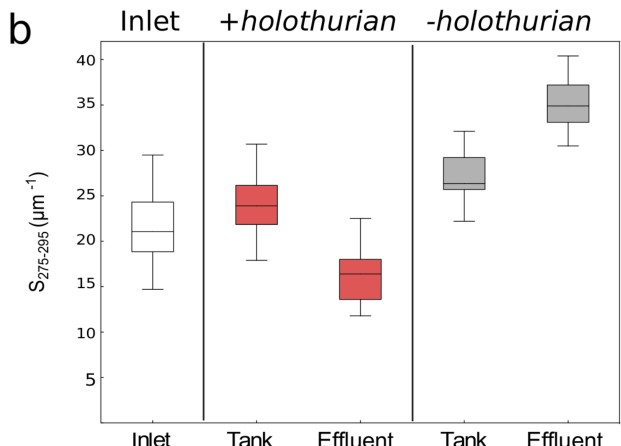

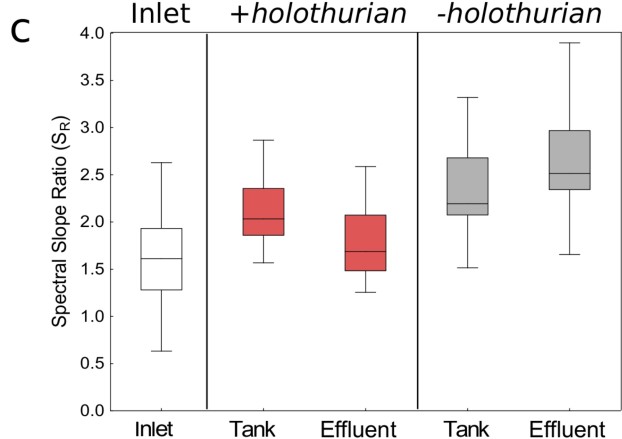

**Figure 3** **Median values (line), the 25–75% percentiles (box), and the non-outlier ranges (whisker) of the optical parameters of chromophoric dissolved organic matter pooling all time-series data.** Values of (A) absorption coefficients at 325 nm ($a_{325}$), (B) spectral slopes from 275 to 295 nm ($S_{275-295}$) and (C) spectral slope ratios ($S_R$) in the inlet water (white box), in the $+holothu$ rian effluent and tank (red boxes) and in the $-holothurian$ effluent and tank (grey boxes).

**Table 2** Results of the analysis of variance (ANOVA) in the three experiments performed to compare the absorption coefficients at 325 nm ($a_{325}$), the absorption coefficients at 443 nm ($a_{443}$), and the spectral slopes from 275 nm to 295 nm ($S_{275-295}$) in the treatments with holothurians (+H) vs. the treatments without holothurians (−H) at initial and final times. Bold means that there is a significant difference between treatments.

| | Experiment # 1 | | Experiment # 2 | | Experiment # 3 | |
|---|---|---|---|---|---|---|
| | *F* | *p*-value | *F* | *p*-value | *F* | *p*-value |
| **Initial time** | | | | | | |
| $a_{325}$ | 5.56 | 0.065 | 0.16 | 0.702 | 0.24 | 0.644 |
| $a_{443}$ | 0.74 | 0.428 | 0.55 | 0.493 | 0.08 | 0.783 |
| $S_{275-295}$ | 1.43 | 0.286 | 0.04 | 0.858 | 2.08 | 0.211 |
| **Final time** | | | | | | |
| $a_{325}$ | **849.74** | **<0.001** | **60.99** | **<0.001** | 3.92 | 0.105 |
| $a_{443}$ | **1980.15** | **<0.001** | 1.70 | 0.249 | 3.14 | 0.137 |
| $S_{275-295}$ | **571.43** | **<0.001** | **506.69** | **<0.001** | **7.89** | **0.038** |

POM) (Fig. 4B). Therefore, POM concentration in the tanks appears to be the main driver of CDOM changes. POM disaggregation into dissolved components is a common process in coastal waters (*He et al., 2016*), particularly under sunny conditions (*Shank et al., 2011*; *Pisani, Yamashita & Jaffé, 2011*). Holothurians consume several components of POM as phytoplankton cells, bacteria, uneaten food, animal feces, and transparent exopolymer particles (*Hudson et al., 2005*; *Slater & Carton, 2009*; *Navarro et al., 2013*; *Yokoyama, 2013*; *Wotton, 2011*). Since holothurians reduce POM concentration by consumption in these tanks (SM Sadeghi-Nassaj, GL Batanero, IP Mazuecos, C Alonso, I Reche, in preparation), we think that POM disaggregation into DOM in the tank with holothurians was significantly lower than in the tank without holothurians where a relevant fraction of POM might have been converted into CDOM. In addition, recently *Brothers, Lee & Nestler (2015)* have demonstrated a direct uptake of free amino acids in several tissues as the respiratory trees, epidermis, and oral tentacles of a sea cucumber species (*Parastichopus californicus*) during the visceral regeneration. It is well known that amino acids such as the tyrosine and tryptophan are able to absorb light in the ultraviolet band (e.g., *Catalá et al., 2013*). Therefore, a direct and selective assimilation of free amino acids by holothurians could also explain a reduction in the $a_{325}$ values and in the spectral slopes.

## Short-term experiments

To corroborate the results obtained in the time-series, we performed three short-term (3 days) experiments in smaller tanks manipulating the presence of holothurians. At the initial time, we did not observe significant differences in the optical parameters between both treatments (+H and −H) indicating the experiments started with identical conditions with the exception of the presence of holothurians (Table 2). However, at the final time after three days, the presence of holothurians (+H treatment) significantly reduced the values of the CDOM absorption coefficients at 325 nm ($a_{325}$) in two out of three experiments (Fig. 5A red bars,

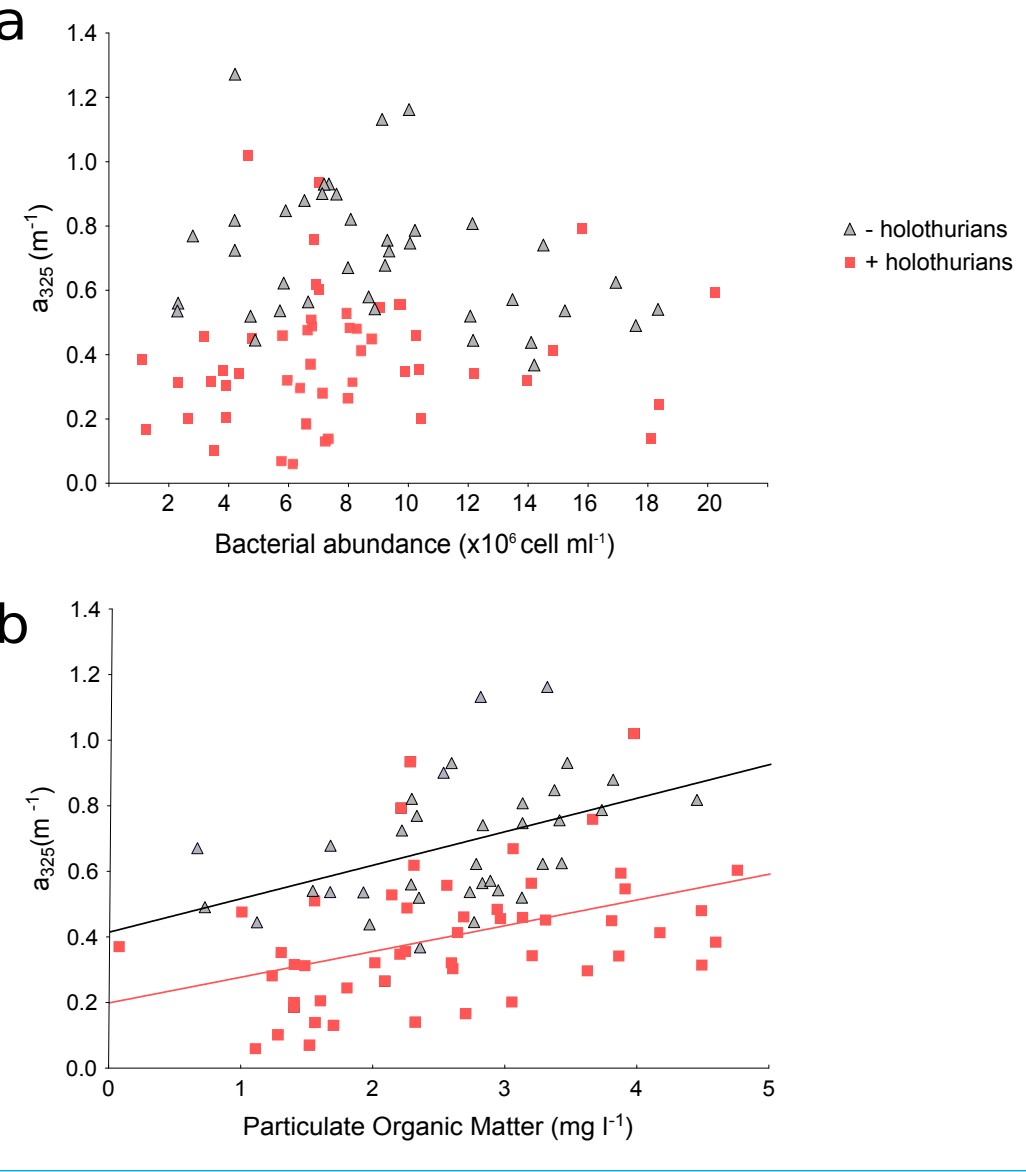

**Figure 4** **Scatterplots of absorption coefficients at 325 nm ($a_{325}$) *vs.* bacterial abundance (A) and particulate organic matter (B) of the time-series data.** Red squares are the values for the +*holothurian* (effluent and tank) waters and grey triangles are the values for the −*holothurian* (effluent and tank) waters. Correlation lines are shown only when are statistically significant ($p < 0.05$).

Table 2) and in one out of three experiments at 443 nm ($a_{443}$) (Fig. 5B red bars, Table 2). In the three experiments, at the final time, the presence of holothurians (+H treatment) reduced significantly the spectral slopes ($S_{275−295}$) (Fig. 5C red bars, Table 2). This variability in the statistical significance for the different optical parameters is related to the inherent higher sensitivity of CDOM as wavelengths are shorter (see Fig. 1).

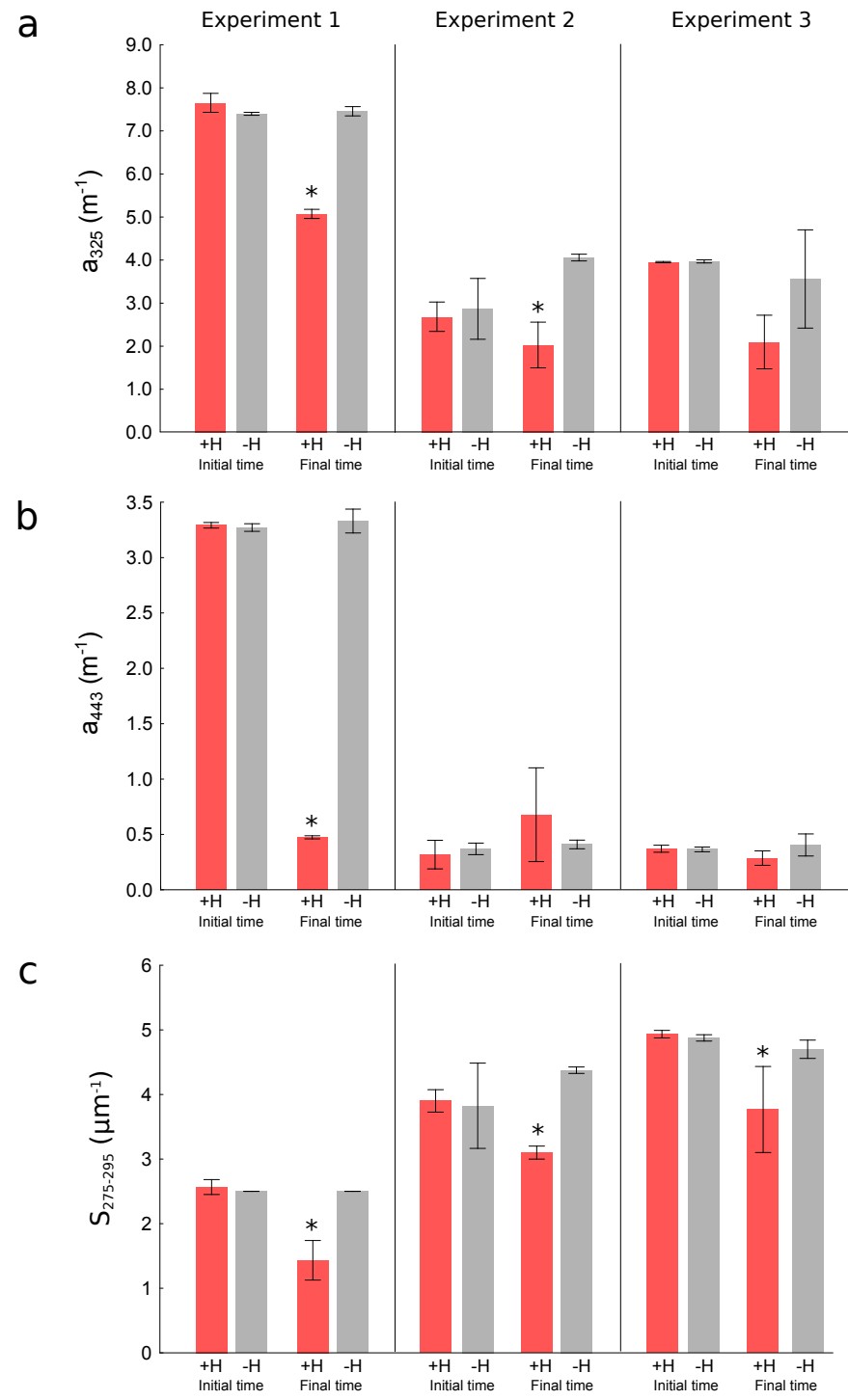

**Figure 5  Changes in optical parameters of chromophoric dissolved organic matter in the three experiments.** Mean (bars) and the standard deviations (whiskers) of the replicates of (A) absorption coefficients at 325 nm ($a_{325}$), (B) absorption coefficients at 443 nm ($a_{443}$), and (C) spectral slopes from 275 to 295 nm ($S_{275-295}$) in the treatments with holothurians (+H) and without holothurians (−H) at the initial and final times. Red bars represent the treatment with holothurians and grey bars represent the treatment without holothurians. Asterisks show the statistically significant results at the final time of the experiments (more details in Table 2).

## CONCLUSIONS

Overall, we found that the presence of holothurians in aquaculture tanks reduces significantly the concentration of CDOM, particularly of compounds with relatively lower molecular size. A plausible mechanism for this reduction is the consumption of particulate organic matter by holothurians reducing its disaggregation into chromophoric dissolved compounds. Complementary, holothurians could also consume directly dissolved compounds as chromophoric amino acids. This fact can affect positively water transparency. Indeed, CDOM optical parameters in the tank with holothurians in the time-series were quite similar to the inlet water both quantitatively (similar absorption coefficients at 325 nm) and qualitatively (similar spectral slopes and ratios) avoiding the increment of color in the waste effluents observed in the tank without holothurians. In offshore aquaculture installations, the presence of holothurians could also affect positively water transparency by reducing light absorption and scattering (*Ibarra, Cembella & Grant, 2012*; *Del Bel Belluz et al., 2016*) as particulate organic matter settle down and they are placed below fish cages. Monitoring CDOM optical properties in aquaculture installations is an easy and inexpensive procedure very sensitive to the changes caused by the extractive species helping in the control of aquaculture waste. Therefore, the use of CDOM probes, for long-term monitoring, or remote sensing for large spatial scales, is a promising research area for the development of a sustainable aquaculture.

All data for CDOM optical parameters and ancillary variables are in supplementary material Tables S1–S5.

## ACKNOWLEDGEMENTS

The authors thank Eulogio Corral Arredondo for help during the samplings, Ana Ortiz with the logistic in the aquaculture tanks, and Gustavo Ortiz Ferrón for his help with flow cytometry for bacterial abundance.

### Funding

This research was supported by the projects CEI BioTic P-BS-46 of the University of Granada and CGL2014–52362R of the Spanish Ministry of Economy and Competitiveness and FEDER funds to Isabel Reche, a PhD fellowship from the Iranian Ministry of Science to Seyed Mohammad Sadeghi-Nassaj, and a postdoctoral contract of the University of Granada to Teresa Serrano Catalá. The funders had no role in study design, data collection and analysis, decision to publish, or preparation of the manuscript.

### Grant Disclosures

The following grant information was disclosed by the authors:
University of Granada: CEI BioTic P-BS-46.
Spanish Ministry of Economy and Competitiveness: CGL2014–52362R.
FEDER.

Iranian Ministry of Science.

## Competing Interests

The authors declare there are no competing interests. Pedro A. Álvarez is founding partner and Research & Development Director of iMare Natural S.L. iMare Natural S.L. is a corporate spin-off of the University of Granada.

## Author Contributions

- Seyed Mohammad Sadeghi-Nassaj performed the experiments, analyzed the data, contributed reagents/materials/analysis tools, wrote the paper, prepared figures and/or tables, reviewed drafts of the paper.
- Teresa S. Catalá analyzed the data, contributed reagents/materials/analysis tools, reviewed drafts of the paper.
- Pedro A. Álvarez conceived and designed the experiments, contributed reagents/materials/analysis tools, reviewed drafts of the paper, use of aquaculture facilities.
- Isabel Reche conceived and designed the experiments, wrote the paper, prepared figures and/or tables, reviewed drafts of the paper.

## Data Availability

The raw data is provided in Supplemental Files.

## Supplemental Information

Supplemental information for this article can be found online at http://dx.doi.org/10.7717/peerj.4344#supplemental-information.

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
