# Peer review of "Sea cucumbers reduce chromophoric dissolved organic matter in aquaculture tanks"

_PeerJ, doi:10.7717/peerj.4344_

## Round 0.1 · original submission · Minor Revisions

The reviewers have evaluated your manuscript that it requires considerable minore revision. Please provide a detailed point-by-point reply to each of the reviewers' comments.

·

Basic reporting

no comment

Experimental design

no comment

Validity of the findings

no comment

Additional comments

This manuscript is excellent, and is the first time I have no recommendations for improvement. The authors executed a straightforward study, comprising a lengthy time series analysis of cnidarian aquaculture tanks, one with and one without holothurians (sea cucumbers) and measured with high frequency cDOM (various metrics, all of which told complementary pieces of the same story of removal of HMW DOM by the holothurians. They then properly designed and executed three replicate experiments to test causality, and found corroborating results. There is no effort to overextrapolate the findings, which is appreciated, but the results are strong and have clear implications for aquaculture management and the ecology of coastal benthic ecosystems.

A few minor grammatical corrections:
L31: “the former being”
L104 “species”
L187-88: Unless otherwise specified model effects (such as repeated experiments as a random factor, etc.), this was probably a t-test done on each experiment?

Reviewer 2 ·

Basic reporting

The main line of this article is clear, and the English is OK. The article structure, figs, tables are good.

However, there are some deficiency in the part of reference:
• The formats of the references are not uniform. Some years are enclosed in brackets, while others are not.

Experimental design

The research of this paper is within the aims and scope of this journal.
The comments that need improving are as follows:

• The equation (2) has some edit error

• Line 171-173. The basis for setting up the number of experimental groups should be explained.

• Line 195-202. The contents of these lines mainly refer to the method of CDOM absorption, and should be transferred to the section “Material and Methods”.

Validity of the findings

• The conclusion of this paper (Line 93-96) seems disobey the common sense and need to be revised. An excessively low water transparency of the aquaculture water is not always good for the aquaculture. It will reduce the photosynthesis of algae. Thus, they will not only produce oxygen but also consume oxygen. In the long run, they will cause ammonia nitrogen, nitrite, hydrogen sulfide and other harmful substances accumulation, resulting in deterioration of water quality. In fact, only if the water transparency was in a reasonable range, the marine products would have a good growth and reproduction. Therefore, the author should supply the data of the water transparency (not just the absorption capacity of dissolved organic matter) to support this conculsion.

• Line 238-241. These contents are lack of sufficient discussion. There is no enough word to explain the reason and mechanism of the holothurians effecting on the molecular weight of DOM in auqaculture water.

• Line 264-267. These sentences are too subjective and need to be further discussed.

Additional comments

In this paper, the author employed sea cucumbers to reduce chromophoric dissolved organic matter in aquaculture tanks. This paper is clear and interesting. However, the paper is lack of deep discussion on the mechanism of sea cucubers reducing CDOM. Thus, this paper is needed to be further modified.

Reviewer 3 ·

Basic reporting

This manuscript describes an interesting observation: that holothurians appear to reduce the CDOM in sea anemone aquacultural effluent. The observations are made biweekly in large tanks over a year and in small tanks in triplicate over 3 days. The results appear to be clear, and the statistical analyses support the conclusion that holothurians do, in fact, have an effect on the transparency of the water. As such, I think this manuscript deserves publication.

There are a number of questions that arise in this new area of research. It would be good for the authors to consider these questions in this paper if possible:

1) It appears that the process of growing A. sulcata in aquacultural tanks produces CDOM. However, the actual source of this CDOM is unknown. Does it come from the feeding of chopped fish? There was no feeding and no increase over initial conditions in the -Holo short term experiments.

2) There are variations in the large tanks over time. What are the possible sources of variability for the influent in the large tanks? Sampling every two weeks does not resolve tidal, diurnal, or temperature effects or many other possible sources of variablility. Is the influent possibly impacted by nearby river flow? Blooms? Changes in coastal current? Is the influent treated, filtered, or settled? Understanding the variability of the influent can help attribute variability of the effluents.

3) For that matter, in the 3 short term experiments, the initial conditions varied quite a bit. Why was the identical influent (starting water) not used?

4) There is an apparent correlation between CDOM and POM in the large tanks. But, I don't see any reduction in POM concentrations in the +Holo compared to the -Holo conditions. There is a stated assumption that Holos reduce POM, and therefore CDOM production…but I do not see the POM reduction. Can this experiment be done or statistics be shown?

5) There is a conclusion that Holos can reduce CDOM in open pens. However, there is no analysis (even back-of-the-envelope calculations) that suggests that Holos can filter a significant amount of water or take up a siginificant amount of POM or CDOM compared to the relevant exchange volumes. Estimates of these parameters would make the potential application more believable than just a trend in a small batch experiment.

Here are some specific questions:

6) TOC is measured and available in the S1-5 tables. However, it is not mentioned in the results or conclusions. It is an important parameter for effluent (maybe more important than CDOM). Is there an effect on TOC by +Holo? If so, state it. If not, this is interesting…as CDOM/TOC ratio is also a measure of CDOM quality.

7) Line 93-04: …presence of holothurians reduced significantly water transparency…" I think you mean increased water transparency….

8) Oveall there are several/many instances that thorough proofreading and editing for English grammar and style are necessary.

Experimental design

9) What is residence time of water in tank? I can calculate, but better if you provide (41 hours)

10) Does food (chopped fish) introduce CDOM to the tank? This would be a simple leaching experiment.

11) Was plastic beaker (plastics are known to sometimes leach CDOM) tested to show that they did not leach CDOM?

Validity of the findings

Results
12) It appears that sea anemones increase the CDOM, and the holo decrease back to influent concentrations. The quality does seem to change a bit (Figure 2) with Holos reducing S275-325.

13) A. sulcata effluent under normal aquaculture condistion is how much volume annually? Compared to coastal exchange? Is it reasonable to think that these tanks may affect coastal water CDOM at maximum aquacultural rates? The authors state this as a possible application of Holos in aquaculture. Some estimates would help make that case.

14) What is the capacity for holothurians to filter water? Can the abundance of these filter the entire large tank before it is replaced by new influent (40 hours?). This will help in interpretation of the impact of Holos.

15) It would be interesting to examine a mass balance for organic carbon in the tanks (small or large). How much mass might go in as food, come out as DOM, be extracted by Holos, etc. Again, this would help in interpreting the data.

16) The authors state that CDOM probes are easy to use to monitor open ocean aquacultural pens. Why did they not use CDOM probes to monitor effluent?. They are suggested for future applications. This might have been helpful to examine variability.

17) Biweekly is not very high resolution. When effluent is higher than influent or vice versa, why? Is there something in the data that might explain this (just after feeding, high temp spike, bloom in influent waters? Etc.

18) Figure 4. More CDOM with more POM. The caption says that no regression line is showen when not significant. However, the data in a (bacteria) and b (POM) do not look that different. Could regression lines with R2 be shown to let the reader decide what is significant?

19) Figure 5. What were differences in conditions of the 3 experiments? Same influent water? Different times? Different animal sources?

Additional comments

Summary
I am still wanting a better mechanistic understanding of the impact of Holos on CDOM. Where does the CDOM come from? How do Holos extract the large MW DOM? How much can the extract? There are all important questions, but maybe it is enough for this first paper to just get the observation out so others can follow up and explore.

---

## Round 0.2 · accepted · Accept

Thank you for the detailed reply to the eidtors's comments.